# Dengue, Zika, and Chikungunya viral circulation and hospitalization rates in Brazil from 2014 to 2019: An ecological study

Julia M. Pescarini[1,2]*, Moreno Rodrigues[3], Enny S. Paixão[1,2], Luciana Cardim[2], Carlos A. A. de Brito[4], Maria da Conceição N. Costa[2,5], Andreia C. Santos[1], Liam Smeeth[1], Maria da Glória Teixeira[2,5], André P. F. de Souza[6‡], Mauricio L. Barreto[2,5‡], Elizabeth B. Brickley[1‡]

1 Faculty of Epidemiology and Population Health, London School of Hygiene & Tropical Medicine, London, United Kingdom, 2 Centro de Integração de Dados e Conhecimento para a Saúde(CIDACS), Gonçalo Moniz Institute, Fundação Oswaldo Cruz, Salvador, Bahia, Brazil, 3 Fundação Oswaldo Cruz, Porto Velho, Rondônia, Brazil, 4 Fundação Oswaldo Cruz, Instituto Aggeu Magalhães, Recife, Pernambuco, Brazil, 5 Instituto de Saúde Coletiva, Universidade Federal da Bahia, Salvador, Brazil, 6 Fundação Getúlio Vargas, São Paulo, São Paulo, Brazil

‡ These authors are co-senior authors on this work.
* Julia.pescarini1@lshtm.ac.uk

**Data Availability Statement:** The final dataset used is available on 10.17632/h7b4nyxmhd.1.

## Abstract

### Background

In addition to their direct pathogenic effects, arthropod-borne (arboviruses) have been hypothesized to indirectly contribute to hospitalizations and death through decompensation of pre-existing comorbidities. Using nationwide data routinely collected from 1 January 2014 to 31 December 2019 in Brazil, we investigated whether local increases in arbovirus notifications were associated with excess hospitalization.

### Methods

We estimated the relative risks for the association between municipality- and state-level increases in arboviral case notifications and age-standardized hospitalization rates (i.e., classified as direct or indirect based on ICD-10 codes) using Bayesian multilevel models with random effects accounting for temporal and geographic correlations. For municipality-level analyses, we excluded municipalities with <200 notifications of a given arbovirus and further adjusted the models for the local Gini Index, Human Development Index, and Family Healthcare Strategy (*Estratégia de Saúde da Família*) coverage. Models for dengue, Zika, and chikungunya were performed separately.

### Results

From 2014 to 2019, Brazil registered 7,566,330 confirmed dengue cases, 159,029 confirmed ZIKV cases, and 433,887 confirmed CHIKV cases. Dengue notifications have an endemic and seasonal pattern, with cases present in 5334 of the 5570 (95.8%) Brazilian municipalities and most (69.5%) registered between February and May. Chikungunya

**Funding:** This work was supported by the British Council Newton Fund (Grant number 527418645 to EBB). The funders had no role in study design, data collection and analysis, decision to publish, or preparation of the manuscript.

**Competing interests:** The authors have declared that no competing interests exist.

notifications followed a similar seasonal pattern to DENV but with a smaller incidence and were restricted to 4390 (78.8%) municipalities. ZIKV was only notified in 2581 (46.3%) municipalities. Increases in dengue and chikungunya notifications were associated with small increases in age-standardized arbovirus-related hospitalizations, but no consistent association was found with all-cause or other specific indirect causes of hospitalization. Zika was associated to increases in hospitalizations by neurological diseases.

## Conclusions

Although we found no clear association between increased incidence of the three arboviruses and excess risks of all-cause or indirect hospitalizations at the municipality- and state-levels, follow-up investigations at the individual-level are warranted to define any potential role of acute arbovirus infection in exacerbating risks of hospitalization from underlying conditions.

### Author summary

Although generally mild and self-limiting, infections with Dengue (DENV), Zika (ZIKV) and Chikungunya (CHIKV) arthropod-borne viruses (arboviruses) can progress to severe complications requiring hospitalization and/or cause death. It has also been hypothesized that arbovirus infections may indirectly lead to hospitalizations or death by decompensating pre-existing comorbidities. We investigated whether increases in notifications of confirmed arbovirus cases in Brazil between 2014 and 2019 were associated with excess hospitalizations from any cause, from arboviruses, or from specific indirect causes (e.g., diabetes). Our findings indicate widespread and seasonal circulation of DENV, ZIKV, and CHIKV in Brazil. In municipalities with at least 200 notifications of a given arbovirus during the study period, increased incidence of confirmed dengue, and chikungunya cases were all associated with small increases in hospitalizations directly caused by arboviruses but were not consistently associated with excess all-cause or indirectly caused hospitalizations. Increases in Zika was associated to increases in hospitalizations by neurological diseases. Nevertheless, this study is limited by the ecological design, and follow-up studies are needed to investigate if arboviruses infections can, at the individual-level, exacerbate pre-existing comorbidities and lead to hospitalizations by indirect causes.

## Background

Dengue (DENV), Zika (ZIKV) and Chikungunya (CHIKV) are arthropod-borne viruses (arboviruses), primarily transmitted by *Aedes spp.* mosquitoes, that increasingly threaten urban and peri-urban populations in the tropical and subtropical areas of low- and middle-income countries. A combination of rapid urban growth, sustained poverty, and warming climates can support unrestrained mosquito proliferation and create the requisite conditions for arboviral outbreaks [1–3].

With almost 85% of its population residing in high-density urbanized settings, stark social inequalities including disparities in housing and sanitation infrastructure [4], and climatic conditions and environmental features (e.g., open water reservoirs) that support

year-round breeding of *Aedes spp*. mosquitoes in much of the country [5], Brazil is uniquely susceptible to arbovirus transmission [6,7]. In 2020 alone, Brazil reported almost 1 million notifications of suspected dengue, 82,419 of suspected chikungunya, and 7,387 of suspected Zika [8]. Nevertheless, this number is likely to be substantially underestimated. Evidence from an enhanced surveillance study investigating DENV underreporting in Salvador, Bahia, between 2009 and 2011 indicates that approximately only 1 in 20 DENV-positive persons who presented to a public emergency unit with acute febrile illness were reported to the Brazilian Information System for Notifiable Diseases (*Sistema de Informação de Agravos de Notificação*, SINAN) [9].

Although asymptomatic infections are common (i.e., estimated in a recent meta-analysis to be 80% for DENV, 50% for ZIKV, and 40% for CHIKV [10]), arboviral infections can lead to disease, hospitalizations, long-term complications, and death. Dengue, Zika, and chikungunya diseases have overlapping clinical presentations with common symptoms including nausea/vomiting, rash, fever, headaches, and joint and muscular pain [11]. Complications of arboviral infections (e.g., Congenital Zika Syndrome or CHIKV-associated arthritis) can have deleterious health and social impacts over the life course, and a subset of infections may trigger severe neurological sequelae (e.g., Guillain-Barré Syndrome) [12] or progress to life-threatening conditions (e.g., severe dengue) [13,14]. Whereas ZIKV and CHIKV-related mortality are rare events, DENV resulted in 554 confirmed deaths in 2020, approximately 0.05% of notified infections in this year [8]. While there is evidence that the risk of severe dengue might be related to non-communicable comorbidities (e.g., diabetes, hypertension, respiratory, cardiovascular or renal diseases) [13–15], whether DENV or other arboviral infections may exacerbate underlying conditions and lead to excess hospitalizations for noncommunicable diseases (NCDs) remains unknown.

Using records of notifications for confirmed dengue, Zika, and chikungunya cases from SINAN and records of hospitalization (*Sistema de Informação Hospitalares*, SIH) in the public Unified Health System (*Sistema Unico de Saude*, SUS) from 1 January 2014 to 31 December 2019, we investigated whether increases in arbovirus notifications rates at the municipality- and state-levels are associated with increased hospitalizations from any cause, from arboviruses, or from specific indirect causes.

## Methods

### Ethics statement

This study was approved by the Ethical Committee of Oswaldo Cruz Foundation (4.756.567) and the London School of Hygiene & Tropical Medicine (25339 /RR/24583).

### Study design and data source

We conducted a municipality- and state-level longitudinal study with data from 1 January 2014 to 31 December 2019. We obtained (i) municipality-level notifications for confirmed dengue, Zika, and chikungunya cases from SINAN; (ii) individual-level hospitalization records from SIH-SUS with International Classification of Diseases 10th revision (ICD-10) coding of causes; (iii) municipality- and state-level annual population estimates from the Brazilian Institute of Geography and Statistics (*Instituto Brasileiro de Geografia e Estatistica*, IBGE) from 2014 to 2019; (iv) municipality-level Human Development Index (M-HDI) and Gini Index for 2010 [16]; and (v) annual municipality-level coverage of the Family Health Strategy (*Estratégia de Saúde da Família*, ESF) from 2014 to 2019 [17].

## Variables and definitions

We used official population estimates [18] for each year from 2014 to 2019 by age in five year strata to calculate (i) the crude monthly incidences of confirmed dengue, Zika, and chikungunya cases per 100,000 inhabitants using as a denominator the population of each municipality or state in a given year, and (ii) the monthly age-standardized hospitalization rates, overall and by ICD-10 cause, standardized by the direct method with the 2010 Brazilian census population as the reference population. To calculate the incidence of dengue, Zika, and chikungunya we only included cases with laboratory or clinical-epidemiological confirmation in SINAN. We considered direct causes of hospitalizations to be those coded for arthropod-borne viral fevers and viral hemorrhagic fevers (ICD-10, A92-A99), and specifically as dengue (A90-A91), chikungunya (A92.0), or Zika (A92.5). We considered indirect causes of hospitalizations to be those caused by a selection of diseases (e.g., diabetes, cerebrovascular diseases, hypertension) and chapters (i.e, chapters III-VII and IX-XIV) that have been previously linked to arboviruses in the literature [13,14,19–22] and confirmed as potentially relevant by two diseases specialists (MGT and CB). See S1 Table for details on the analyzed ICD-10 codes.

## Study population

For the descriptive analyses, we included all municipalities and states. For the statistical analyses at the municipality-level, we restricted the datasets to municipalities that identified at least 200 confirmed cases of dengue, Zika, and chikungunya during the study period. To reflect the initiation of Zika and chikungunya registrations in Brazil, we restricted our analysis for Zika to 1 August 2015 to 31 December 2019 and our analysis for chikungunya to 1 January 2015 to 31 December 2019.

## Analysis

We estimated the incidence of confirmed dengue, Zika, and chikungunya cases overall and within each state, considering as the denominator the population of municipalities with at least one case of each disease. Our primary analysis estimated whether dengue, Zika, and chikungunya incidences are associated with changes in hospitalizations rates overall and by direct (i.e., related to arboviruses) and indirect (i.e., related to other causes of hospitalizations) subcauses. We estimated the relative risks (RR) and 95% Credible Intervals (95% CrI) by adjusting a Bayesian multilevel model with two random effects. We considered as random effects a random walk process of order 1 (i.e., assuming that time t is correlated to time t-1) to account for temporal correlation in months, and an independent and identically distributed (IID) Gaussian random effect to account for geographic clustering at the municipality-level. As higher prevalences of chronic NCDs are concentrated among individuals living in poorer socioeconomic quintiles [23], who also might be more exposed to arbovirus infection, we also adjusted the models using fixed effects for the 2010 M-HD, 2010 Gini Index, and the annual coverage of the ESF. Each variable used as fixed-effect was attributed a weak informative prior. We obtained the precision for the hyperparameters using the logarithm of logGamma distribution and fitted the model using Integrated Nested Laplace Approximation (INLA) [24,25]. As many Brazilian municipalities are small and do not have tertiary care, we implemented State-level models to test if municipality- and state-level estimates led to similar results. For state-level models, HDI and ESF coverage are not available at the state-level and, therefore, were not included as confounding variables. To test if changes in hospitalizations were more likely to occur within the next calendar month or two, for each model, we performed a sensitivity analysis including parameters related to one- or two-month delays between notifications and hospitalizations. We used the deviance information criterion (DIC) to evaluate the performance

of all models [26]. Indirect causes of hospitalizations that had both the median and mean incidence distribution equal to zero were not included in the analysis (See S2 and S3 Tables). For each outcome model, the model was only implemented for municipalities that had at least 50% of studied months with at least one hospitalization for that specific outcome. All models were implemented in R 4.0.3 (R Development Core Team, 2020) using the package *INLA*. Maps were created in Python using the package *Folium* and graphs were created in *R* using *ggplot*. The final dataset used is available on 10.17632/h7b4nyxmhd.1.

## Results

From 1 January 2014 to 31 December 2019, the Brazilian health surveillance system registered 7,566,330 confirmed dengue cases, 159,029 confirmed Zika cases, and 433,887 of confirmed chikungunya cases in SINAN. While dengue cases were registered in almost all Brazilian municipalities (5334/5570, 95.8%), only 1791 municipalities (32.2%) registered at least 1 case of Zika, and 2523 (45.3%) registered at least one case of chikungunya. During the study period, notifications of dengue per 100,000 were higher than for Zika and chikungunya and covered a greater geographic distribtion–Zika and chikungunya affected fewer municipalities but incurred a relatively high incidence within the subset of municipalities that registered at least one case of each of the diseases (See Fig 1A–1C and S4 Table). Whereas dengue notifications were mostly observed in municipalities from Southeast and Central-west Brazilian States, Zika notifications have been concentrated in the North, Northeast and Central-west, and chikungunya incidence has been more spread across the North and Northeast regions (See S1 Fig).

When investigating the association between arbovirus notifications and direct and indirect causes of hospitalizations in municipalities that detected at least 200 cases of each disease, we included 2366 municipalities (44.4% of the 5334 with ≥1 case) for dengue, 99 municipalities (5.5% of the 1791 with ≥1 case) for Zika, and 217 municipalities (8.6% of the 2523 with ≥1 case) for chikungunya. Dengue notifications were associated with a small increase in dengue-related hospitalizations (RR 1.0029, 95% Crl 1.0027–1.003) and similar increases in non-haemorragic (RR 1.0029, 95% Crl 1.0027–1.003) and haemorrhagic dengue hospitalizations (RR 1.0029, 95% Crl 1.0019–1.0034) (Table 1). Zika notifications were not associated with dengue (RR 1.0002, 95% Crl 0.9986–1.0019) or arthropod-borne viral fevers and viral haemorrhagic fevers hospitalizations (RR 1.0076, 95% Crl 0.9441–1.082). Chikungunya notifications, on the other hand, were associated with overall increases in dengue (RR 1.0044, 95% Crl 1.0037–1.0051), and arthropod-borne viral fevers and viral haemorrhagic fevers hospitalizations (RR 1.0067, 95%Crl 1.0034–1.0111). At the state level, increases in dengue and chikungunya notifications were also associated with increases in dengue (dengue RR 1.0053, 95% Crl 1.0047–1.0059; chikungunya RR 1.0061, 95% Crl 1.0035–1.009) and chikungunya hospitalizations (dengue RR 1.0048, 95% Crl 1.0027–1.0072; chikungunya RR 1.0206, 95% Crl 1.0115–1.0310).

At the municipality level, we found no increases in all-cause hospitalization rates associated to arboviruses notifications. Zika incidence was associated with small increases in hospitalizations for endocrine, nutritional, and metabolic diseases (RR 1.0003, 95% Crl 1.0001–1.0004), diabetes (RR 1.0004, 95% Crl 1.0002–1.0006) and inflammatory polyneuropathy (including Guillain-Barré) (RR 1.0082, 95% Crl 1.0033–1.0133) (see Table 2). Chikungunya and dengue incidence were not associated with any indirect cause of hospitalization. At the state-level, only associations between Zika and hospitalization from inflammatory polyneuropathy as an indirect cause was consistent with the analysis at the municipality-level (Table 3). Finally, in sensitivity analyses investigating potential one- or two-month delays in the association between increases in dengue, zika and chikungunya notifications and hospitalizations, we found similar point estimates for all the analyses (S5 to S10 Tables).

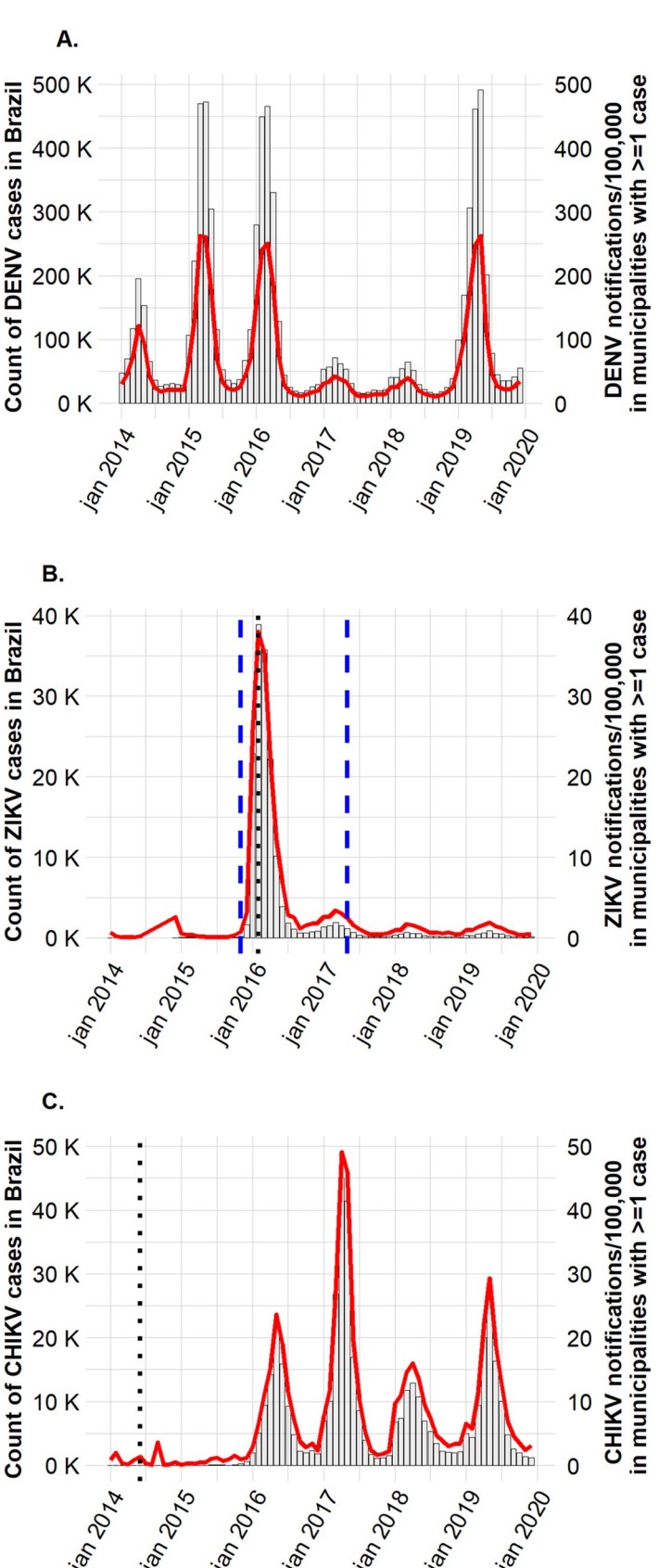

**Fig 1.** Number of (A) DENV, (B) ZIKV and (C) CHIKV case notifications (bars, main Y axys) and notifications per 100,000 (red line, secondary Y axys) in Brazilian municipalities with at least one notification between 2014 and 2019. *Please note that the scales differ between the three graphs. The dotted black lines in graphs B and C mark the start of ZIKV and CHIKV mandatory notification in Brazil. The dashed blue lines in graph B mark the start and the end of Zika Public Health emergency.*

## Discussion

This nationwide ecological study found that between 2014 and 2019, dengue and chikungunya notifications in Brazil were associated with a small increase in arbovirus-related hospitalizations (i.e., arthropod-borne viral fevers and viral haemorrhagic fevers, dengue, and chikungunya). Zika was consistently associated with increases in hospitalizations by inflammatory polyneuropathy (including Guillain-Barré), but not by other indirect causes. In contrast, increases in dengue and chikungunya notifications were not consistently associated with increases in all-cause or indirectly caused hospitalizations at the municipality- or state-levels.

The association between notification rates of confirmed arbovirus cases and arbovirus-related hospitalizations is expected, but still of high public health relevance. Previous research suggests the direct and indirect cost of dengue ambulatory and hospitalization treatments in Brazil are an estimated 447,000,000 USD per year, with costs related to deaths of approximately 65,000,000 USD per year [27]. Our finding of an association between arbovirus notifications and arbovirus-related hospitalizations likely reflects local environmental and social conditions that may support the proliferation of competent vector populations and lead to high forces of infection for different arboviruses (and serotypes) within a given community over time.

The finding of associations of increased dengue and chikungunya notifications with chikungunya-related hospitalizations and associations between increased chikungunya notifications with dengue-related hospitalizations indicate temporal and spatial co-circulation of arboviruses within a given community and may also reflect locally enhanced arbovirus surveillance and diagnostic capacity. While current evidence suggests that co-infections between ZIKV, CHIKV, and DENV do not substantively increase risks of arbovirus disease severity [28], repeated flavivirus infections are a particular concern for dengue hospitalizations.

**Table 1. Changes of monthly age-standardized hospitalization rates directly associated with DENV, ZIKV and CHIKV cases incidence at municipality and at State level.**

| Arbovirus-related hospitalizations | DENV incidence | ZIKV incidence | CHIKV incidence |
|---|---|---|---|
| | RR (95% CrI) | RR (95% CrI) | RR (95% CrI) |
| *Municipality level[1]* | (N = 2,366 municipalities) | (N = 99 municipalities) | (N = 217 municipalities) |
| Dengue (A90-A91) | 1.0029 (1.0027–1.003) | 1.0002 (0.9986–1.0019) | 1.0044 (1.0037–1.0051) |
| Dengue non-haemorragic (A90) | 1.0029 (1.0027–1.003) | 1.0001 (0.9985–1.002) | 1.0044 (1.0037–1.0051) |
| Dengue haemorragic (A91) | 1.0027 (1.0019–1.0034) | 1.0032 (0.9949–1.0124) | 1.0028 (1.0004–1.0057) |
| Arthropod-borne viral fevers and viral haemorrhagic fevers (A92-A99) | 1.0002 (0.9995–1.0009) | 1.0076 (0.9441–1.082) | 1.0067 (1.0034–1.0111) |
| *State level* | (N = 27 states) | (N = 27 states) | (N = 27 states) |
| Dengue (A90-A91) | 1.0053 (1.0047–1.0059) | 1.0052 (0.9994–1.0118) | 1.0061 (1.0035–1.009) |
| Dengue non-haemorragic (A90) | 1.0052 (1.0047–1.0058) | 1.0052 (0.9994–1.0118) | 1.0064 (1.0037–1.0093) |
| Dengue haemorragic (A91) | 1.0057 (1.0048–1.0066) | 1.0032 (0.9944–1.0132) | 1.0022 (0.9995–1.0052) |
| Arthropod-borne viral fevers and viral haemorrhagic fevers (A92-A99) | 1.0021 (1.0012–1.003) | 1.0030 (0.9960–1.0111) | 1.0283 (1.0212–1.0359) |
| Chikungunya virus disease (A92.5) | 1.0048 (1.0027–1.0072) | 1.0237 (0.9929–1.0589) | 1.0206 (1.0115–1.031) |

[1] Analysis restricted to municipalities with at least 200 cases of the diseases. RR adjusted for the Human Development Index, Gini Index and coverage of the family health strategy (FHS).

NA[2] We did not calculate not statistical significant excess cases of arboviral infections.

**Table 2. Changes of monthly age-standardized hospitalization rates indirectly associated with Dengue, Zika and Chikungunya incidence in municipalities with at least 200 cases of the diseases.**

| | DENV incidence | ZIKV incidence | CHIKV incidence |
| --- | --- | --- | --- |
| | RR[1] (95% CrI) | RR[1] (95% CrI) | RR[1] (95% CrI) |
| *Municipality level*[1] | (N = 2,366 municipalities) | (N = 99 municipalities) | (N = 217 municipalities) |
| *All causes* | 1 (1–1.0001) | 1.0001 (1–1.0002) | 1 (1–1.0001) |
| *By chapter* | | | |
| Diseases of the blood and blood-forming organs and certain disorders involving the immune mechanism (D50-D89) | 1.0001 (1–1.0001) | 1.0002 (1–1.0004) | 1.0003 (1.0001–1.0004) |
| Endocrine, nutritional and metabolic diseases (E00-E89) | 1.0001 (1–1.0001) | **1.0003 (1.0001–1.0004)** | 1 (0.9999–1.0001) |
| Diseases of the circulatory system (I00-I99) | 1 (1–1) | 1.0001 (1–1.0002) | 1 (1–1.0001) |
| Mental and behavioural disorders (F01-F99) | 1 (1–1) | 1.0003 (1–1.0005) | 0.9999 (0.9997–1) |
| Diseases of the nervous system (G00-G99) | 1 (1–1) | 1.0001 (0.9999–1.0003) | 1 (0.9999–1.0001) |
| Diseases of the eye and adnexa (H00-H59) | 0.9999 (0.9999–1) | 0.9995 (0.9988–1.0001) | 0.9998 (0.9994–1.0003) |
| Diseases of the respiratory system (J00-J99) | 1 (1–1) | 1 (0.9999–1.0002) | 1 (1–1.0001) |
| Diseases of the digestive system (K00-K95) | 1 (1–1) | 1.0001 (1–1.0002) | 1 (0.9999–1) |
| Diseases of the skin and subcutaneous tissue (L00-L99) | 1 (1–1) | 1.0001 (0.9999–1.0003) | 1 (0.9999–1.0001) |
| Diseases of the musculoskeletal system and connective tissue (M00-M99) | 1 (1–1) | 1 (0.9998–1.0002) | 0.9999 (0.9997–1) |
| Diseases of the genitourinary system (N00-N99) | 1 (1–1) | 1.0001 (1–1.0003) | 0.9999 (0.9999–1) |
| *By causes* | | | |
| Diabetes mellitus (E10-E13) | 1 (1–1.0001) | **1.0004 (1.0002–1.0006)** | 1.0001 (0.9999–1.0002) |
| Cerebrovascular diseases (I60-I69) | 1 (1–1.0001) | 1.0002 (1–1.0003) | 1 (0.9999–1.0001) |
| Hypertensive diseases (I10-I15) | 1 (1–1.0001) | 1.0002 (0.9999–1.0005) | 0.9999 (0.9997–1.0001) |
| Ischemic heart diseases (I20-I25) | 1.0001 (1–1.0001) | 1 (0.9998–1.0002) | 1 (0.9998–1.0001) |
| Inflammatory diseases of the central nervous system (G00-G09) | 1 (0.9999–1) | 1.0024 (0.9994–1.0052) | 1.0002 (0.9995–1.001) |
| Encephalitis, myelitis and encephalomyelitis; Encephalitis, myelitis and encephalomyelitis in diseases classified elsewhere (G04-G05) | 0.9999 (0.9995–1.0002) | 1.006 (0.9937–1.0181) | 0.9995 (0.9984–1.0007) |
| Sequelae of inflammatory diseases of central nervous system (G09) | 1 (0.9999–1.0001) | 1.0041 (0.9975–1.0096) | 1.0057 (0.998–1.0135) |
| Acute myocarditis (I40) | 0.9987 (0.9968–1.0002) | 0.9995 (0.9796–1.0198) | 1.0068 (0.9927–1.0204) |
| Arthropathies (M00-M25) | 1 (1–1) | 0.9999 (0.9995–1.0002) | 0.9998 (0.9996–1.0001) |
| Inflammatory polyneuropathy (including Guillain-Barré) (G61) | 1.0002 (0.9999–1.0004) | **1.0082 (1.0033–1.0133)** | 1.0003 (0.9989–1.0017) |
| Pregnancy with abortive outcome (O00-O08) | 1 (1–1) | 1.0001 (0.9999–1.0003) | 0.9999 (0.9998–1) |

[1] Adjusted for the Human Development Index, Gini Index and coverage of the family health Strategy.

Although primary DENV infections are thought to confer some degree of cross-protection against heterologous DENVs in the short-term, previous immune experience with a heterologous DENV serotype is considered, in the long-term, to be a key risk factor for developing severe dengue that requires hospitalization upon secondary infection [29,30]. Limited evidence suggests that previous ZIKV infection may similarly facilitate antibody dependent

**Table 3. Changes of monthly age-standardized hospitalization rates associated with Dengue, Zika and Chikungunya incidence in the 27 Brazilian States.**

| | DENV incidence | ZIKV incidence | CHIKV incidence |
|---|---|---|---|
| | RR (95% CrI) | RR (95% CrI) | RR (95% CrI) |
| *All causes* | 1.0001 (1–1.0002) | 1 (0.9987–1.0015) | 1.0003 (0.9998–1.0009) |
| *By chapter* | | | |
| Diseases of the blood and blood-forming organs and certain disorders involving the immune mechanism (D50-D89) | 1.0001 (1–1.0002) | 1.0003 (0.9988–1.0019) | 1.0004 (0.9998–1.0011) |
| Endocrine, nutritional and metabolic diseases (E00-E89) | 1 (0.9999–1.0002) | 1.0011 (0.9997–1.0026) | 1.0001 (0.9995–1.0007) |
| Diseases of the circulatory system (I00-I99) | 1 (0.9999–1.0002) | 0.9999 (0.9985–1.0014) | 1.0003 (0.9998–1.0009) |
| Mental and behavioural disorders (F01-F99) | 1.0001 (0.9999–1.0003) | 1.0007 (0.9987–1.0028) | 1.0005 (0.9996–1.0014) |
| Diseases of the nervous system (G00-G99) | 1 (0.9999–1.0002) | 0.9993 (0.9977–1.0009) | 1 (0.9994–1.0007) |
| Diseases of the eye and adnexa (H00-H59) | 1.0001 (0.9998–1.0004) | 1.0005 (0.9978–1.0034) | 1.0005 (0.999–1.002) |
| Diseases of the respiratory system (J00-J99) | 1.0002 (1–1.0004) | 1.0007 (0.999–1.0025) | 1.0005 (0.9999–1.0012) |
| Diseases of the digestive system (K00-K95) | 1 (0.9999–1.0002) | 1.0001 (0.9987–1.0016) | 1.0003 (0.9997–1.0009) |
| Diseases of the skin and subcutaneous tissue (L00-L99) | 1 (0.9998–1.0001) | 1.0009 (0.9993–1.0025) | **1.0008 (1.0001–1.0014)** |
| Diseases of the musculoskeletal system and connective tissue (M00-M99) | 1.0001 (0.9999–1.0002) | 1.0001 (0.9984–1.0019) | 1.0001 (0.9994–1.0008) |
| Diseases of the genitourinary system (N00-N99) | 1 (0.9999–1.0002) | 1 (0.9985–1.0014) | 1.0003 (0.9997–1.0009) |
| *By causes* | | | |
| Diabetes mellitus (E10-E13) | 1 (0.9999–1.0002) | 1.0014 (0.9997–1.0031) | 1.0007 (1–1.0014) |
| Cerebrovascular diseases (I60-I69) | 1 (0.9999–1.0002) | 1.0001 (0.9986–1.0018) | 1.0005 (0.9999–1.0012) |
| Hypertensive diseases (I10-I15) | 1 (0.9999–1.0002) | 0.9993 (0.9976–1.001) | 1.0001 (0.9994–1.0008) |
| Ischemic heart diseases (I20-I25) | 1 (0.9998–1.0001) | 0.9991 (0.9976–1.0007) | 0.9999 (0.9993–1.0005) |
| Inflammatory diseases of the central nervous system (G00-G09) | 1.0001 (0.9999–1.0004) | 0.9991 (0.9962–1.0021) | 0.9998 (0.9987–1.0009) |
| Encephalitis, myelitis and encephalomyelitis; Encephalitis, myelitis and encephalomyelitis in diseases classified elsewhere (G04-G05) | 0.9999 (0.9995–1.0004) | 0.9995 (0.9921–1.0068) | 0.9991 (0.9976–1.0007) |
| Sequelae of inflammatory diseases of central nervous system (G09) | 1 (0.9994–1.0006) | 1.0027 (0.9935–1.0125) | 0.9961 (0.9892–1.0033) |
| Acute myocarditis (I40) | 0.9998 (0.9992–1.0003) | 1.002 (0.9956–1.0078) | 1 (0.9963–1.0032) |
| Arthropathies (M00-M25) | 0.9999 (0.9997–1.0001) | 0.9999 (0.998–1.0019) | 0.9996 (0.9988–1.0004) |
| Inflammatory polyneuropathy (including Guillain-Barré) (G61) | **1.0004 (1.0001–1.0007)** | **1.0054 (1.0025–1.0084)** | 1.0006 (0.9993–1.002) |
| Pregnancy with abortive outcome (O00-O08) | 0.9999 (0.9998–1) | 0.9997 (0.9982–1.0012) | 0.9996 (0.999–1.0002) |

*Model presented poor fit

enhancement of subsequent DENV infection, increasing risks of severe dengue [31]. Overall, our findings reinforce the value of reducing community-level arbovirus exposure over time through integrated initiatives to improve *Aedes* mosquito control, living conditions and sanitation, and access to personal protective measures, such as topical repellants (reviewed in [32]).

As Zika is a generally mild disease and rarely leads to hospitalizations and deaths, we could not estimate associations between increases in arboviruses notifications and ZIKV-related hospitalizations [8]. Even among children and adolescents, evidence from a recent systematic review indicates postnatal ZIKV infections cause neurological complications in less than 1% of cases [33]. Nevertheless, the fact that the notifications of Zika were restricted to a small number of municipalites and that notification rates are relatively small, small increases in hospitalization due to indirect Zika complications might not have been observed at the ecological level.

Although Zika and dengue are known to have the potential to cause neurological complications [12,34], we only found consistent association between Zika incidence and hospitalizations by Inflammatory diseases of the central nervous system or by inflammatory polyneuropathies, which include Guillain-Barré Syndrome. In endemic countries, hospitalized dengue patients have been found to present encephalopathies, encephalitis, neuromuscular complications, and neuro-ophthalmic involvement (Reviewed in [34]). ZIKV infection has been associated with severe complications, such as Guillain-Barré Syndrome in adults [12,20], but the incidence of these complications among all ZIKV infected patients is very small. In a prospective observational study carried out in Recife, Brazil, on patients presenting with neurological complications possibly related to arboviral infections, Guillain-Barré was the most prevalent complication among the ZIKV-infected patients [12]. Differently, central nervous system (CNS) diseases such as myelitis were more prevalent among CHIKV-infected patients [12].

Although Zika was associated with Endocrine, nutritional and metabolic diseases (which includes diabetes) and diabetes itself, overall there was a lack of consistent association between arboviruses increases and increases in hospitalization from all causes or other specific indirect causes, including by chronic non-communicable diseases, such as cerebrovascular diseases, hypertensive or ischaemic heart diseases. Even though we found no association between increases in dengue notifications and indirect causes of hospitalizations, previous studies provide compelling evidence that dengue disease in individuals with comorbidities, such as diabetes, stroke, hypertension, or respiratory, cardiovascular, or renal diseases, are more likely to progress to severe dengue [13–15] and death [19]. In addition, we found no increases in indirect hospitalizations associated to increases in chikungunya notifications, but previous studies have shown that 2 to 10% of chikungunya patients can present arthralgia and/or myalgia [21], and that the infection could exacerbate preexisting conditions and lead to hospitalization [22]. Acute signs of CHIKV infection (i.e., fever, arthralgia and myalgia) are often accompanied by chronic effects in the liver, muscle, joints and remote lymphoid organs that can lead to hospitalization and that may persist for weeks to years [21]. Similarly to dengue, a study conducted in the Réunion Island has shown that hypertension and underlying respiratory or cardiological conditions are independent risk factors for developing a severe case of chikungunya disease (i.e., as defined as a case that required maintenance of at least one vital function) and that could lead to death [35]. However, even though CHIKV rarely leads to severe cases and hospitalizations [36], several studies have found higher mortality during epidemics of CHIKV [37–40]. A study conducted in Brazil looking at excess deaths during the CHIKV pandemic months in 2016 compared to previous years found a 21% increase in deaths in the State of Pernambuco, 17% in Rio Grande do Norte and from 5–15% in different regions of Bahia associated to that year chikungunya epidemic [37]. Studies using the same methodology found smaller increases in mortality associated with 2014 and 2016 chikungunya epidemics were also

observed in Puerto Rico, India, Guadelupe, Martinique, and the Dominican Republic, where the epidemics reached smaller incidences than in the Brazilian state of Pernambuco [37–40].

Our study analyzed nationwide data of confirmed arbovirus cases notified in Brazil and used all public hospitalizations from SUS, which comprised the majority of hospitalization registered in the country. Nevertheless, this study is subject to limitations inherent to the ecological study design—although we found no consistent association between higher arbovirus incidences at the municipality level and increased hospitalization rates by indirect causes, we cannot rule out the possibility that individuals infected by arboviral infection will have higher risk of discompensating from a previous comorbidity and being hospitalized by that disease. Similarly, we are unable to unable to discern whether a given individual experienced repeated arbovirus infections and that would increase their risk of hospitalization by both direct (arboviral-related) or indirect causes. The small proportion of arboviral cases that lead to hospitalization and the potential excess cases of hospitalizations that could occur in the peak periods of diseases transmission might be very sensitive to underreporting of cases. Dengue registries are subject to high levels of underreporting, which are likely to be higher in periods of the year when the disease is less prevalent [9], which could lead to the insufficient capacity of finding associations between an increase in notifications of arboviruses in general and admissions to hospital, especially at the municipality- or state-level. In the case of Zika, previous studies have also shown that nearly 10% of cases were reported in pregnant persons (34), indicating bias in disease reporting patterns.

## Conclusions

In this study, we found evidence that increases in dengue, and chikungunya notifications were associated with small increases in age-standardized arbovirus-related hospitalizations, but no consistent association was found with all-cause or other specific indirect causes of hospitalization. We also found that increases in hospitalizations by neurological diseases have been associated with increases in Zika notifications. We showed that strong associations between arboviral infections and increases in complications that lead to hospitalization can be detected at ecological municipal or state level analysis, but further investigations at the individual-level are necessary to understand if arboviral infections have the potential to exacerbate individual risks of hospitalization from underlying conditions, such as chronic diseases. Understanding these associations could improve management of patients, lead to the investigation of arboviral infections among patients hospitalized by chronic conditions in endemic areas for arboviruses, and reinforce the need to reduce community-level arbovirus exposure.

## Supporting information

**S1 Table. Studied chapters and subcauses of hospitalizations.**
(DOCX)

**S2 Table. Mean yearly notifications per 100,000 inhabitants during the studied considering as the denominator all municipalities and considering only those who in a given month had at least one case of the disease.**
(DOCX)

**S3 Table. Descriptive of monthly dengue, zika and chikungunya incidence and age-standardized hospitalization rates in the 5570 Brazilian municipalities from 2014 to 2019.**
(DOCX)

**S4 Table. Descriptive of monthly dengue, zika and chikungunya incidence and age-standardized hospitalization rates in the 27 Brazilian states.**
(DOCX)

**S5 Table. Changes of monthly age-standardized hospitalization rates associated with Dengue in municipalities with at least 200 cases of the disease considering an effect at the same month, and with 1 or 2 months delay.**
(DOCX)

**S6 Table. Changes of monthly age-standardized hospitalization rates associated with Zika in municipalities with at least 200 case of the disease considering an effect at the same month, and with 1 or 2 months delay.**
(DOCX)

**S7 Table. Changes of monthly age-standardized hospitalization rates associated with chikungunya in municipalities with at least 200 case of the disease considering an effect at the same month, and with 1 or 2 months delay.**
(DOCX)

**S8 Table. Changes of monthly age-standardized hospitalization rates associated with Dengue in the 27 Brazilian States considering an effect at the same month, and with 1 or 2 months delay.**
(DOCX)

**S9 Table. Changes of monthly age-standardized hospitalization rates associated with Zika in the 27 Brazilian States considering an effect at the same month, and with 1 or 2 months delay.**
(DOCX)

**S10 Table. Changes of monthly age-standardized hospitalization rates associated with chikungunya in the 27 Brazilian States considering an effect at the same month, and with 1 or 2 months delay.**
(DOCX)

**S1 Fig.** Monthly Dengue (A), Zika (B) and Chikungunya (C) incidence in Brazil per 100,000 (notifications per population per month) from 2014 to 2019. *Map built using the libray folium (https://python-visualization.github.io/folium/) on Leaflet (https://leafletjs.com/) and with layer provided by OpenStreetMap (https://www.openstreetmap.org/#map = 4/-15.33/-47.29). All libraries are open and freely available.*
(MP4)

## Author Contributions

**Conceptualization:** Julia M. Pescarini, Enny S. Paixão, Maria da Conceição N. Costa, Andreia C. Santos, Liam Smeeth, Maria da Glória Teixeira, André P. F. de Souza, Mauricio L. Barreto, Elizabeth B. Brickley.

**Data curation:** Julia M. Pescarini, Moreno Rodrigues, Luciana Cardim.

**Formal analysis:** Julia M. Pescarini, Moreno Rodrigues.

**Funding acquisition:** Julia M. Pescarini, Enny S. Paixão, Andreia C. Santos, Liam Smeeth, André P. F. de Souza, Mauricio L. Barreto, Elizabeth B. Brickley.

**Investigation:** Julia M. Pescarini, Moreno Rodrigues, Enny S. Paixão, Carlos A. A. de Brito, Maria da Conceição N. Costa, Andreia C. Santos, Liam Smeeth, Maria da Glória Teixeira, André P. F. de Souza, Mauricio L. Barreto, Elizabeth B. Brickley.

**Methodology:** Julia M. Pescarini, Moreno Rodrigues, Carlos A. A. de Brito, Maria da Conceição N. Costa, Maria da Glória Teixeira, Elizabeth B. Brickley.

**Project administration:** Mauricio L. Barreto, Elizabeth B. Brickley.

**Supervision:** Mauricio L. Barreto, Elizabeth B. Brickley.

**Validation:** Carlos A. A. de Brito, Maria da Conceição N. Costa, Maria da Glória Teixeira.

**Visualization:** Julia M. Pescarini, Moreno Rodrigues, Elizabeth B. Brickley.

**Writing – original draft:** Julia M. Pescarini, Moreno Rodrigues, Luciana Cardim.

**Writing – review & editing:** Julia M. Pescarini, Enny S. Paixão, Luciana Cardim, Carlos A. A. de Brito, Maria da Conceição N. Costa, Andreia C. Santos, Liam Smeeth, Maria da Glória Teixeira, André P. F. de Souza, Mauricio L. Barreto, Elizabeth B. Brickley.

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
