## [Decision Letter · Decision Letter 0]

22 Jun 2022

Dear Miss Pescarini,

We are pleased to inform you that your manuscript 'Dengue, Zika, and Chikungunya viral circulation and hospitalization rates in Brazil from 2014 to 2019: an ecological study.' has been provisionally accepted for publication in PLOS Neglected Tropical Diseases.

Best regards,

Nigel Beebe, PhD

Deputy Editor

Nigel Beebe

Deputy Editor

Reviewer's Responses to Questions

**Key Review Criteria Required for Acceptance?**

**Methods**

-Are the objectives of the study clearly articulated with a clear testable hypothesis stated?

-Is the study design appropriate to address the stated objectives?

-Is the population clearly described and appropriate for the hypothesis being tested?

-Is the sample size sufficient to ensure adequate power to address the hypothesis being tested?

-Were correct statistical analysis used to support conclusions?

-Are there concerns about ethical or regulatory requirements being met?

Reviewer #1: Methods were very adequately describe and the study design excellent. Sample size is enormous, providing very high power to the anaklyses. There are no ethical concerns noted.

Reviewer #2: In the manuscript, the objective of the study is clearly established and it is hypothesized that arboviruses contribute indirectly to hospitalizations and deaths through the decompensation of pre-existing comorbidities.

The ecological approach of the study on information on arboviruses (dengue, chikungunya and zika) in all geographic units of Brazil (municipalities and states) is adequate to answer the research question. The Bayesian model is applied for the analysis of this information and the relative risk is used as a measure of association. The research team adequately restricts the period of analysis for Chikungunya and Zika, considering the epidemiology of these arboviruses in Brazil

The study was approved by an ethics committee

**Results**

-Does the analysis presented match the analysis plan?

-Are the results clearly and completely presented?

-Are the figures (Tables, Images) of sufficient quality for clarity?

Reviewer #1: Excellent job of presenting results and the figures are reflective and summarize the results.

Reviewer #2: The results are presented in main and supplementary tables due to the large amount of information that the study team wanted to present. The graphics are repeated, it is suggested to consider the ones with the best resolution

**Conclusions**

-Are the conclusions supported by the data presented?

-Are the limitations of analysis clearly described?

-Do the authors discuss how these data can be helpful to advance our understanding of the topic under study?

-Is public health relevance addressed?

Reviewer #1: Although the base hypothesis was not supported by the analyses performed, this paper provides exceptionally valuable insights into medical consequences in an are with intense arbovirus transmission.

Reviewer #2: The conclusions fit the results. The limitations are characteristic of this type of study approach and are adequately mentioned in the manuscript, establishing the need for future studies on this topic.

**Editorial and Data Presentation Modifications?**

Reviewer #1: The paper is very well written and I have not suggestions to improve it other than one strange capitalized "Endocrine".

Reviewer #2: (No Response)

**Summary and General Comments**

Reviewer #1: I consider this an outstanding manuscript that investigated the intriguing hypothesis that endemic arbovirus infections would lead to greater all cause morbidity from such disorders as diabetes. Despite the fact that the hypothesis was not supported by the analyses, this remains and excellent and valuable contribution.

Reviewer #2: The co-circulation of arboviruses is a public health problem that produces a significant economic burden in endemic countries such as Brazil. The hypothesis raised by the researchers continues to be the focus of discussion and this study with an ecological approach provides initial evidence and serves as a guide for future studies with an individual approach.

PLOS authors have the option to publish the peer review history of their article (what does this mean?). If published, this will include your full peer review and any attached files.

Reviewer #1: No

Reviewer #2: No

---

## [Editor Report · Acceptance letter]

22 Jul 2022

Dear Miss Pescarini,

We are delighted to inform you that your manuscript, "Dengue, Zika, and Chikungunya viral circulation and hospitalization rates in Brazil from 2014 to 2019: an ecological study.," has been formally accepted for publication in PLOS Neglected Tropical Diseases.

Best regards,

Shaden Kamhawi

co-Editor-in-Chief

Paul Brindley

co-Editor-in-Chief
